# The Role of Health Behaviors in Quality of Life: A Longitudinal Study of Patients with Colorectal Cancer

**DOI:** 10.3390/ijerph20075416

**Published:** 2023-04-06

**Authors:** Jaroslaw Ocalewski, Michał Jankowski, Wojciech Zegarski, Arkadiusz Migdalski, Krzysztof Buczkowski

**Affiliations:** 1Department of Health Psychology, Faculty of Psychology, Kazimierz Wielki University, 85-064 Bydgoszcz, Poland; jareko@ukw.edu.pl; 2Department of Surgical Oncology, L. Rydygier Collegium Medicum in Bydgoszcz, Nicolaus Copernicus University, 85-094 Toruń, Poland; michaljankowski@post.pl (M.J.); zegarskiw@co.bydgoszcz.pl (W.Z.); 3Department of Surgical Oncology, Oncology Center—Professor Franciszek Lukaszczyk Memorial Hospital, Romanowskiej, 85-796 Bydgoszcz, Poland; 4Department of Vascular Surgery and Angiology, L. Rydygier Collegium Medicum in Bydgoszcz, Nicolaus Copernicus University, 85-094 Torun, Poland; armigos@wp.pl; 5Department of Family Medicine, L. Rydygier Collegium Medicum in Bydgoszcz, Nicolaus Copernicus University, 85-094 Torun, Poland

**Keywords:** quality of life, colorectal cancer, health behaviors, smoking, alcohol, physical activity

## Abstract

Colorectal cancer (CRC) is the third most common malignancy and the second most common cancer-related cause of death worldwide. CRC incidence depends, in part, on the health behaviors that make up an individual’s lifestyle. We aimed to assess the influence of health behaviors and quality of life (QoL) among patients with CRC receiving surgical treatment. In this single-center questionnaire study, 151 patients were surveyed 1 week before and 6 months after colorectal procedures (laparoscopic hemicolectomy, low rectal anterior resection, abdominoperineal resection, and others). This study demonstrated a significant decrease in alcohol consumption and physical activity following the execution of colorectal procedures. No statistically significant changes were observed in smoking or the consumption of healthy food. Global QoL did not change significantly; however, a decrease in physical and role-related functioning was observed. Significant improvements in emotional functioning were also observed. A detailed analysis showed that physical and social functioning were related to smoking, the consumption of healthy food, physical activity, and additional therapies. Emotional functioning was related to smoking, the consumption of healthy food, and complementary treatments. Six months following an operation, it was also dependent on alcohol intake. Physical functioning was the area that decreased the most in the six months after colorectal tumor surgery compared to the period before surgery. Health behaviors such as cessation of smoking, engagement in physical activity, and the consumption of healthy food contributed to a higher quality of life among patients prior to resecting colorectal cancer and six months after the procedure. Patients who received adjuvant/neoadjuvant therapy had a lower quality of life than patients who did not receive this type of therapy. The kind of surgery (laparoscopic hemicolectomy, lower anterior rectum resection, or abdominoperineal rectum resection) was not related to QoL six months after surgery.

## 1. Introduction

According to the World Health Organization, Quality of Life (QoL) is an individual’s perception of their position in life regarding their culture and systems of values and in relation to their expectations, goals, standards, and concerns [1]. Quality of life encompasses the entirety of the physical, psychological, social, and functional aspects of human experiences and behaviors that are experienced by the person concerned [2]. In studies conducted on chronically ill patients, the term health-related quality of life is used. QoL is a subjective assessment of a person that includes both positive and negative aspects of life. QoL explores the complexity of human health through aspects of an individual’s physical, mental, and social functioning. The physical components of QoL include indices related to health/illness status, age, physical pain, and life expectancy. The mental aspects include anxiety, depression, and a patient’s thoughts or beliefs about their illness. The corresponding social features are explored through status, achievements, resources, ability to perform social roles, and marginalization. Together, these aspects offer a picture that can be used to describe a patient.

Colorectal cancer (CRC) is the third most frequent cancer in the world and the second leading cause of death related to cancer. The highest rates of morbidity are in Australia and Europe (more than 40/100,000 among men), and the highest mortality rate is in Eastern Europe (more than 20/100,000 among men) [3,4]. It is likely that this situation will intensify in the upcoming decades, particularly in wealthy countries [4]. In the last four decades, among the male population of Poland, CRC-related trends in mortality have increased, with a slight decrease observed in 2018 [5]. Among Polish women, the growing CRC mortality trend halted in the mid-1990s; since then, a downward trend has been observed [5].

CRC incidence depends, in part, on the health behaviors that make up an individual’s lifestyle [6]. Eating red meat, drinking alcohol, and obesity are associated with a much more frequent occurrence of CRC [7,8,9]. Conversely, regular physical activity reduces CRC morbidity. In addition to lifestyle factors, some patients also have a genetic predisposition to CRC [10,11].

Enhanced recovery after surgery (ERAS) is a multimodal protocol for perioperative patient care [12]. ERAS includes recommendations for patients regarding healthy behaviors to engage in when preparing for surgery, including stopping smoking, decreasing alcohol consumption, following a balanced diet, and maintaining adequate physical activity (30–60 min walks) [13]. Adherence to these recommendations significantly decreases hospitalization times and lowers the rate of postoperative complications and readmissions [14].

Quality of life refers to an individual’s subjective assessment of their life situation. In the case of cancer patients, their health condition is an important factor of their quality of life. The symptoms of the disease, such as pain, have a negative impact on a patient’s psychological well-being and quality of life [15]. Studies indicate that coping strategies mediate the relationship between pain and well-being [16,17]. Coping can involve emotional strategies such as relaxation, distraction, or meditation as well as behavioral strategies such as engaging in physical activity, adopting a healthy diet, or using stimulants such as cigarettes or alcohol.

Studies have shown that health behaviors are related to QoL. In patients with CRC, an improvement in QoL related to physical functioning is observed as a result of increased physical activity and adherence to a proper diet [18]. Health behaviors that favor increased QoL include the consumption of vegetables and fruits [19,20] and engagement in physical activity [21,22]. Studies also emphasize the negative impacts of smoking and drinking alcohol on QoL [20,23]. The use of adjuvant therapies such as chemotherapy can also be negatively related to a patient’s quality of life. One study showed that patients with colorectal cancer who were treated with 5-fluorouracil, leucovorin, and oxaliplatin (FOLFOX) chemotherapy showed symptoms of peripheral neuropathy, which directly reduced their quality of life compared to patients who had not undergone chemotherapy [24]. Significantly worse mental and physical quality-of-life scores were observed among patients with CRC who received chemotherapy [25]. In addition, patients with rectal cancer often undergo perioperative radiotherapy, which significantly, particularly in combination with surgery, affects their QoL, causing issues such as difficulty passing stool or the requirement for a stoma [26].

Despite the current literature related to QoL among colorectal patients, there is a lack of research assessing QoL before and after CRC surgery. We seek to fill this gap using a research plan that accounts for several recommendations (the cessation of smoking, reducing alcohol consumption, the consumption of healthy food, and increasing physical activity) of the ERAS protocol and highlights their importance for CRC patients’ QoL.

This study aims to investigate the correlation between health behaviors and quality of life both before and six months after surgery among CRC patients who underwent colorectal tumor removal surgery.

## 2. Method

### 2.1. Materials and Methods

#### Sampling

The participants were selected from patients referred to the Oncology Center of Lukaszczyk Memorial Hospital (a high-volume center with over 400 colorectal resections per year), Poland, for CRC resection. Using purposive sampling, study participation was offered to all CRC resection patients staying at the Department of Surgical Oncology. Inclusion criteria were as follows: (1) diagnosis of CRC in one of the following locations: colon (ICD 10: C18), rectosigmoid junction (ICD 10: C19), rectum (ICD 10: C20), or anal canal (ICD 10: C21); (2) qualifying for surgical CRC tumor removal (by means of laparoscopic hemicolectomy, lower anterior resection of the rectum, abdominoperineal resection of the rectum, or other surgical procedure); and (3) being of the age of 18 years or older. Exclusion criteria included a history of any type of cancer and a risk of malnutrition (BMI < 18.5 kg/m^2^) in the period before CRC surgery.

For the first measurement, before surgery (T1), 151 people participated (172 people met criteria but 21 people refused to participate) (Figure 1). For the second measurement, half of a year after surgery (T2), 105 patients from among the respondents surveyed in T1 participated. A total of 21 people refused to participate in the T2 study, and 25 people could not be contacted (they did not answer phone calls, no follow-up appointment was recorded in the oncology center, or death was reported).

All participants supplied written informed consent before the project’s initiation. The study was conducted from 1 June 2018 to 30 April 2019. Participation was voluntary and free of charge.

### 2.2. Measures

#### 2.2.1. Health-Related Behaviors

For this study, the authors limited the assessment of behaviors related to health to those that the ERAS protocol indicates are the most significant for surgery.

Smoking, alcohol intake, physical activity, and diet were assessed in the same way as in the Health Status of Population in Poland 2014 study conducted by the Central Statistical Office as part of European Health Interview Survey (EHIS) [27]. In addition to the questions about the mean number of cigarettes smoked and the amount of alcohol consumed included in the EHIS survey, the phrase “after falling ill (in the last month)” was added, and the scale of responses was refined to six possible answers. Questions about the consumption of products that were included in the EHIS were limited to vegetables and fruits, whereas in this study, we included fish and wholegrain bread. In the case of physical activity, its intensity was not differentiated according to the season, and the question about the frequency of performing various physical activities lasting at least 30 min was narrowed down.

To identify smoking status and amounts, the participants were asked to determine the number of cigarettes smoked per day, for which non-smokers were asked to indicate a sore of “0”.

The intake of alcohol was assessed according to the frequency of consumption (1—never; 2—from one to three times a month; 3—once per week; 4—a few times per week; 5—once a day; and 6—a few times daily) and the size of a portion (single portions were defined as 250 mL of beer, 100 mL of wine, and 30 mL of vodka or other spirits). A weekly alcohol intake indicator was created by multiplying the frequency of alcohol consumption by the amount.

Healthy food products, including fruits, vegetables [28,29], fish, and wholegrain bread [30,31], were chosen based on research about factors that may increase the risk of cancer or decrease the number of postoperative complications a patient may experience. Satisfactory reliability of the “intake of healthy food” scale was obtained using a Cronbach α = 0.60.

The mean age of the CRC patients was approximately 65 years and the preliminary study showed that a small fraction of them engaged in sports. Thus, physical activity was assessed by selecting sports activities and actions related to daily activities; specifically, five options were chosen: cycling, walking, engaging in housework that requires physical activity, gardening, and other kinds of physical activity. The surveyed patients were expected to determine the frequency with which they performed each type of activity per week, for which it was assumed that a single event lasted at least 30 min (Cronbach α = 0.66).

#### 2.2.2. Quality of Life

Quality of life was assessed using the EORTC QLQ-C30 questionnaire (European Organization for Research and Treatment of Cancer Quality of Life Questionnaire) [32]. The QLQ-C30 consists of 30 questions describing global quality of life conditioned as impacted by health, functional status, and severity of symptoms related to the disease. The scales assessing functional conditions refer to physical functioning, fulfillment of social roles, emotional functioning, memory and concentration, and social functioning. The symptom scales include fatigue, nausea and vomiting, and pain. Scale scores range from 0 to 100. A higher score on a functional status scale indicates better functioning, while a higher value on the symptom scale suggests greater severity. Aronson et al. obtained Cronbach α values ranging from 0.52 to 0.89 [32].

#### 2.2.3. Demographics and Medical Data

We collected the following demographic and medical data on all patients: gender; age; place of residence; marital and legal status; education; date of cancer diagnosis (month and year); location of tumor, including the colon (ICD 10: C18), rectosigmoid junction (ICD 10: C19), rectum (ICD 10: C20), and anal canal (ICD 10: C21); the extent of cancer’s spread (according to tumor, node, and metastasis classification—TNM); the type of therapy performed before the surgery (chemotherapy—CTx; radiotherapy—RT); the kind of surgery and treatment applied (laparoscopic hemicolectomy—LH; low rectal anterior resection—LAR; abdominoperineal resection—APR; or other); and the kind of postsurgical treatment applied (chemotherapy, radiotherapy, or other).

### 2.3. Study Procedure

The measurements were collected in a longitudinal study conducted a week before surgery (T1) and six months after surgery (T2). The selection of an interval of six months after surgery was based on clinical observations that indicated that this period is when supplementary chemotherapy is completed and the patients’ physical functioning relatively improves. Every patient received an information leaflet based on the ERAS protocol prior to measurement. At T1, respondents were asked to report the health-related behaviors they engaged in the month before surgery. At T2, patients reported if they had made any changes or experienced any increases/decreases regarding health-related behaviors in the last month. Data were collected using pen-and-paper questionnaires.

### 2.4. Participants

As previously mentioned, the T1 measurement involved 151 patients, while the T2 measurements involved 105 patients (Figure 1). Both groups were similar in terms of age (in T1: M_age_ = 64.89 and SD_age_
*=* 10.14; in T2: M_age_ = 64.30 and SD_age_ = 10.51), gender (T1 vs. T2: men—66.23% vs. 67.62% and women—33.77% vs. 32.38%; χ^2^ = 0.05 and *p* = 0.816), place of residence (T1 vs. T2: city—66.89% vs. 61.90% and country—33.11% vs. 38.10%; χ^2^ = 0.67 and *p* = 0.412), marital status (T1 vs. T2: single—4.64% vs. 4.76%; married—76.16% vs. 78.10%; widowed—15.89% vs. 15.24%; and divorced—3.31% vs. 1.90%), education (T1 vs. T2: primary—14.57% vs. 11.43%; vocational—34.44% vs. 35.24%; secondary—35.76% vs. 35.24%; and higher—15.23% vs. 18.09%), cancer location (T1 vs. T2: colon—33.77% vs. 35.24%; rectosigmoid junction—11.26% vs. 12.38%; rectum and anal canal—48.34% vs. 45.71%; colon or rectum of uncertain or unknown location—6.62% vs. 6.67%), type of surgery performed (T1 vs. T2: laparoscopic hemicolectomy—31.79% vs. 31.43%; low rectal anterior resection—45.70% vs. 49.52%; abdominoperineal resection—22.52% vs. 19.05%), and the extent of the spread of cancer (T1 vs. T2:0—5.96% vs. 4.76%; I—19.87% vs. 20.00%; II—27.81% vs. 31.43%; III—44.37% vs. 49.90% and IV—1.99% vs. 1.90%) were obtained in T1 and T2 (Table 1).

The authors conducted a dropout analysis of QoL among patients who did not participate in T2. The level of QoL in T1 (M = 61.41; SD = 20.59) among the patients who did not participate in T2 was not significantly different (*p* = *0*.510) than that among all participants at T1 (M = 63.58; SD = 19.21). This permitted a comparison of the results of T1 and T2.

### 2.5. Statistical Analyses

Statistical analyses were performed using the STATISTICA 13 software. The level of significance, *p*, was set at 0.05. Cronbach’s alpha was used to assess the internal consistency of the health-related behaviors. Repeated ANOVA was applied to assess each behavior related to health. The size of the object was also assessed (*η^2^*). To determine the differences in QoL resulting from the type of surgery (laparoscopic hemicolectomy, low rectal anterior resection, or abdominoperineal resection), the Kruskal–Wallis test was used. QoL analysis was also performed with respect to the implementation of a stoma; for this purpose, the analysis of comparisons of two separate groups was carried out using the Mann–Whitney U test. To demonstrate the relationship between health behaviors and physical, emotional, and social functioning, we conducted a set of multivariate linear regression analyses of T1 and T2. To avoid collinearity, we performed a test using the Variance Inflation Factor (VIF). The acquired level did not exceed VIF > 10 in any of these models; thus, it can be inferred that there were weak correlations between the independent variables (additionally, the VIF for individual predictors did not exceed 2) [33].

### 2.6. Ethical Approval

The study was carried out in compliance with the Declaration of Helsinki, and the protocol was approved by the Bioethics Committee of Collegium Medicum Nicolaus Copernicus University in Torun, Poland (Protocol KB345/2017). Written informed consent was gathered from all participants before they participated in the study.

## 3. Results

The resulting levels of health-promoting behaviors (number of cigarettes smoked daily, weekly alcohol consumption, frequency of consuming healthy food, and number of minutes spent engaging in physical activity per week) and quality-of-life outcomes during T1 and T2 are presented in Table 2.

The number of cigarettes smoked, although decreased, did not change statistically significantly in T2 (M = 1.48; SD = 5.40) compared to T1 (M = 2.12; SD = 5.65) (*F*(1,104) = 1.83; *p* = 0.180; η^2^ = 0.02) (Table 2). Alcohol intake decreased significantly at T2 (*M* = 0.80; *SD* = 2.40) compared to T1 (*M* = 1.20; *SD* = 4.84) (*F*(1,104) = 2.68; *p* = 0.012; η^2^ = 0.06). There were no changes in the consumption of healthy food between T2 (*M* = 15.69; *SD* = 2.64) and T1 (*M* = 15.01; *SD* = 3.25). There was a decrease in physical activity in T2 vs. T1 (*M* = 319.43 and *SD* = 206.71 vs. *M* = 388.61 and *SD* = 308.28) (*F*(1,104) = 6.55; *p* = 0.012; η^2^ = 0.06). Global QoL did not change significantly, although an upward trend was visible (T2 vs. T1: *M* = 67.67 and *SD* = 19.62 vs. *M* = 63.58 and *SD* = 19.21 (*F*(1,104) = 1.73; *p* = 0.192; η2 = 0.02). However, there was a decrease in physical functioning (T2 vs. T1: *M* = 80.44 and *SD* = 19.13 vs. *M* = 83.84 and *SD* = 16.17 (*F*(1,104) = 10.16; *p* = 0.002.; η^2^ = 0.09) and role functioning (T2 vs. T1: *M* = 82.06 and *SD* = 24.32 vs. *M* = 85.76 and *SD* = 23.76) (*F*(1,104) = 4.86; *p* = 0.029; η2 = 0.05). An increase in emotional functioning was noticed at T2 (*M* = 89.06; *SD* = 12.68) compared to T1 (*M* = 75.37; *SD* = 23.44) (*F*(1,104) = 36.29; *p* < 0.001; η2 = 0.26). There were no statistically significant changes in cognitive functioning in T2 vs. T1 (*M* = 86.98 and *SD* = 15.67 vs. *M* = 84.55 and *SD* = 20.01) (*F*(1,104) = 2.09; *p* = 0.151; η2 = 0.02), social functioning (*M* = 88.57 and *SD* = 17.50 vs. *M* = 86.98 and *SD* = 17.64) (*F*(1,104) = 0.32; *p* = 0.571; η^2^ < 0.01), fatigue (*M* = 24.23 and *SD* = 21.23 vs. *M* = 27.08 and *SD* = 24.16) (*F*(1,104) = 0.58; *p* = 0.446; η^2^ = 0.01), and pain (*M* = 14.29 and *SD* = 21.49 vs. *M* = 19.09 and *SD* = 24.97) (*F*(1,104) = 3.00; *p* = 0.086; η^2^ = 0.03).

The next level of analysis determined differences in QoL resulting from the kind of surgery performed (laparoscopic hemicolectomy, low rectal anterior resection, or abdominoperineal resection) (Table 3). The following results were obtained: global QoL—H(2,105) = 0.31 and *p* = 0.857; physical functioning—H(2,105) = 0.971 and *p* = 0.615; role functioning—*H*(2,105) = 1.41 and *p* = 0.494; emotional functioning—*H*(2.105) = 1.54 and *p* = 0.463; cognitive functioning—*H*(2,105) = 0.41 and *p* = 0.816; and social functioning—*H*(2.105) = 1.77 and *p* = 0.413. QoL analysis was also performed with regard to the application of a stoma. For this purpose, an analysis consisting of the comparison of two independent groups was performed. There were no differences in both global QoL and particular types of QoL (*n* = 105: patients with applied stoma, *n* = 67; patients without application of stoma, *n* = 38): global QoL—*U* = 1122.50 and *p* = 0.317; physical functioning—*U* = 1272.50 and *p* = 1.000; role functioning—*U* = 1229.00 and *p* = 0.314; emotional functioning—*U* = 1228.00 and *p* = 0.767; cognitive functioning—*U* = 1165.50 and *p* = 0.476; and social functioning—*U* = 1263.00 and *p* = 0.949 (Table 3).

To verify the relationship between health behaviors and physical, emotional, and social functioning, we carried out a complete set of multivariate linear regressions for T1 and T2. The explanatory variables were as follows: the number of cigarettes smoked, alcohol consumption, frequency of consumption of healthy food, physical activity, and neoadjuvant treatment (T1)/adjuvant treatment (T2).

The multivariate linear regression model for QoL—physical functioning (Table 4) revealed the following statistically significant independent variables: number of cigarettes (T1: *β* = −0.18; *p* = 0.020; T2: *β* = −0.13; *p* = 0.20), frequency of consuming healthy food (T1: *β* = 0.22; *p* = 0.006; T2: *β* = 0.19; *p* = 0.046), physical activity (T1: *β* = 0.26; *p* < 0.001; T2: *β* = 0.41; *p* < 0.001), and complementary treatment (neoadjuvant therapy/T1: *β* = −0.18; *p* = 0.012; adjuvant therapy/T1: *β* = −0.19; *p* = 0.039) (Table 4). Alcohol intake was not a significant variable with respect to QoL—social functioning (T1: *β* = 0.08; *p* = 0.266; T2: *β* = *0*.06; *p* = 0.566). The T1 model explained 18% of the variance in QoL physical functioning [*F*(5,142) = 7.52; *p* < 0.001], while the T2 model explained 22% [*F*(5,93) = 6.90; *p* < 0.001]. Physical activity at T1 and T2 explained 10% and 19% of the variance in QoL—physical functioning, respectively.

The Variance Inflation Factor was 5.34 in T1 and 5.70 in T2, thus indicating acceptable collinearity for the analyzed variables.

The multivariate linear regression model for QoL—social functioning (Table 5) revealed the following statistically significant independent variables: number of cigarettes (T1: *β* = −0.36; *p* < 0.001; T2: *β* = −0.34; *p* < 0.001), frequency of healthy food intake (T1: *β* = 0.33; *p* < 0.001; T2: *β* = 0.31; *p* < 0.001), physical activity (T1: *β* = 0.29; *p* = 0.010; T2: *β* = 0.21; *p* = 0.021), and complementary treatment (neoadjuvant therapy/T1: *β* = −0.18; *p* = *0*.018; adjuvant therapy/T1: *β* = −0.02; *p* = 0.823) (Table 4). Alcohol intake was not a significant variable with respect to QoL—social functioning (T1: *β* = 0.10; *p* = 0.159; T2: *β* = 0.14; *p* = 0.154). The T1 model explained 25% of the variance in QoL—social functioning [*F*(5,138) = 10.52; *p* < 0.001], while the T2 model explained 20% [*F*(5,94) = 6.10; *p* < 0.001]. The number of cigarettes smoked and the frequency of consuming healthy food in T1 and T2 explained 11% and 9% as well as 8% and 10%, respectively, of the variance in QoL—social functioning.

The Variance Inflation Factor was equal to 5.25 in T1 and 5.44 in T2, thus indicating acceptable collinearity for the analyzed variables.

The multivariate linear regression model for QoL—emotional functioning (Table 6) revealed the following statistically significant independent variables: the number of cigarettes (T1: *β* = −0.38; *p* < 0.001; T2: *β* = −0.31; *p* = 0.003), the intake frequency of pro-health products (T1: *β* = 0.29; *p* < 0.001; T2: *β* = 0.21; *p* = 0.029), and complementary treatment (neoadjuvant therapy/T1: *β* = −0.19; *p* = 0.011; adjuvant therapy/T1: *β* = −0.21; *p* = 0.003) (Table 5). Alcohol intake was only a significant variable for QoL—emotional functioning in T2, where *β* = 0.20 and *p* = 0.047. Physical activity was not a significant variable with respect to QoL—emotional functioning (T1: *β* = 0.05; *p* = 0.480; T2: *β* = 0.01; *p* = 0.957).

The T1 model explained 20% of the variance in QoL—emotional functioning [*F*(5,145) = 8.31; *p* < 0.001] and the T2 model explained 12% [*F*(5,97) = 3.82; *p* < 0.001]. The intake frequency of pro-health products and the number of cigarettes smoked in T1 explained 9% of the variance for each in T1 and 5% and 3% of the variance in T2, respectively, with respect to QoL—emotional functioning. Alcohol consumption was associated with QoL—emotional functioning in T2. The Variance Inflation Factor was equal to 5.25 in T1 and 5.55 in T2, indicating acceptable collinearity for the analyzed variables.

## 4. Discussion

This study compared healthy behaviors one week before and six months after surgery for CRC. No statistically significant changes were observed with respect to smoking or the consumption of healthy food. However, this study demonstrated significant decreases in alcohol consumption and physical activity.

Global QoL did not change significantly; however, significant decreases in physical and role functioning were observed. In addition, significant improvements in emotional functioning were observed.

A detailed analysis showed that physical and social functioning were related to smoking, the consumption of healthy food, physical activity, and complementary treatment. Emotional functioning was related to smoking, the consumption of healthy food, and complementary treatments; six months after an operation, it was also dependent on alcohol intake.

In accordance with the ERAS protocol, which contains recommendations regarding preparation for surgery, we analyzed cigarette smoking and showed that cigarette smoking lowers the physical, social, and emotional functioning scores related to QoL. Likewise, a British study conducted among patients diagnosed with CRC within the last five years proved that non-smoking is beneficial with respect to global QoL [20]. Research has confirmed that, even in the general population, cigarette smokers present lower QoL than non-smokers and the cessation of smoking improves QoL [34,35,36]. Many mechanisms impact this effect, but within the scope of physical performance, a decrease in oxygen caused by smoking reduces physical endurance, thus hampering patients’ ability to perform well in sports and engage in everyday activities such as walking up stairs. The impact of smoking on emotional functioning can be explained by the nicotine contained in cigarettes, which penetrates the central nervous system, causing sensations such as relaxation, increased concentration, or an improved mood. Conversely, cigarette smokers’ limitation or abandonment of smoking is associated with negative feelings such as increased anxiety, mood deterioration, and, possibly, increased fear [37,38]. This may partially explain why there is a lower level of emotional functioning among smokers who try to limit the number of cigarettes they smoke.

The positive relationship between alcohol consumption and emotional functioning is interesting (as increased alcohol consumption increases the level of emotional functioning). To explain this phenomenon, one should consider the number of portions consumed weekly. The results regarding the alcohol intake of patients with CRC showed that substance intake was reduced at T2 (1.20 parts weekly before surgery and 0.80 parts per week after it). It can be assumed that the patients followed their physician’s advice in accordance with the ERAS protocol. Six months after surgery, this beneficial change was maintained. Such an observation could also result from more frequent contact with relatives and, therefore, the consumption of smaller amounts of alcohol. Research conducted among colorectal cancer survivors has shown that the vast majority of respondents are abstinent or that if they drink alcohol, they only do so in small amounts [39]. Alcohol has a strong effect on the brain and nervous system. After drinking alcohol, people become more relaxed and confident in their abilities, make new acquaintances easier, and are less inhibited from expressing their thoughts. This affordance increases the emotional functioning of patients who consume moderate amounts of alcohol (> 1 and <14 units for women; > 1 and <21 units for men) [6].

No changes in statistical significance were found regarding the consumption of healthy food before or six months after surgery. However, we found a positive relationship between eating healthy food (such as vegetables, fruits, fish, and whole-grain bread) and QoL-related physical, social, and emotional functioning. The most commonly eaten products were fruits and vegetables; however, the results obtained were averages, indicating that these items were only consumed a few times per week. Research conducted among people who underwent surgery for CRC has shown increased consumption of milk, vegetables, and fruits two years after diagnosis [40]. Fruits and vegetables are valuable sources of vitamins, micro- and macro-elements, and substances that support antioxidant effects [40]. Their regular consumption supports the excretory system and intestinal peristalsis. This directly translates into improved physical functioning and the possibility of being active in various social spheres. In Poland, nutritional problems mainly concern the deficiency of dietary products such as fruits and vegetables. This problem is more frequently experienced by older people in adverse financial situations than by those who are economically stable [27]. Among working people, it was also noticed that they consume meals quickly and irregularly; consume more fatty, sweet, and salty snacks rich in calories; and often skip breakfast.

The relationship between consuming healthy food and improved emotional functioning can be characterized as the “food-mood connection mechanism.” This suggests that a person who consumes a particular food product combines inputs from numerous receptors (e.g., taste, smell, visual, and auditory) with their current emotional state during a single experience. Through repeated experiences, an individual’s association between healthy foods and positive emotional states is strengthened [41].

The results regarding levels of physical activity showed a reduction in the level of physical activity after CRC surgery compared to that before surgery. It seems to be a common problem that physical activity decreases after being diagnosed with cancer, as confirmed by other researchers [42,43]. In our study, physical activity was significantly associated with both physical and social functioning. It is worth noting that physical activity was not a predictor of emotional functioning in either T1 or T2 assessments. Perhaps this lack of association is due to the first measurement being taken a week before surgery when the anticipatory anxiety level of waiting patients might be higher. In addition, the six-month period after surgery may be so short that patients retain a high degree of fear for cancer recurrence (FCR), which does not have a positive effect on emotional functioning. FCR among patients who underwent CRC was relatively stable during therapy; however, it decreased only five years after diagnosis [28,44]. In Poland, the level of activity is determined mainly by the age of the individual. After the age of 29, the level of physical activity decreases quite rapidly [25].

This study also revealed a relationship between adjuvant/neoadjuvant therapy and decreased QoL. Perioperative treatment (preoperative/postoperative radio- and chemotherapy) plays a crucial role in the contemporary treatment of CRC patients but might have an important impact on QoL. Preoperative radiotherapy is a risk factor for low anterior resection syndrome, which significantly reduces quality of life [45]. Similarly, radiotherapy significantly increases sexual dysfunction [46] and impairs urination, particularly with respect to distal tumors treated with abdominoperineal resection [47].

The patients who had undergone additional adjuvant treatment had lower levels of emotional functioning than those who had not.

It was expected that patients’ quality of life would depend on the type of surgery, as this determines the course of convalescence and differentiates the risk of complications and treatment consequences, such as stoma formation. Patients with CRC often suffer from a number of uncomfortable symptoms, such as rectal bleeding and gastrointestinal obstruction, which reduce their quality of life and may be associated with a higher probability of receiving a temporary or even permanent stoma. The occurrence and severity of these symptoms increase when the tumor is located more distally. The type of resection (abdominoperineal resection or anterior resection) undoubtedly affects patients’ quality of life but is determined by the location of the disease. Notably, although maintaining a patient’s physiological route of defecation improves their quality of life, effective radical treatment has the same effect regardless of the type of surgery [48]. It is also interesting that the type of procedure was not associated with an increase in behaviors related to health during the preoperative and postoperative periods. This seems to have been because the patients’ health behaviors and QoL were measured only six months after surgery. Only after postoperative wounds are fully healed, which is a duration that varies depending on the type of surgical procedure, can patients display a higher level of physical activity.

The stabilization of a disease helps increase QoL. The time since surgery may be a predictor of adaptation and improved mental functioning. This thesis was confirmed by the studies conducted by Akechi et al., which included four groups of patients categorized by the time of diagnosis to disease recurrence, arranged as follows: up to 1 year, 1–3 years, over 3 years, and people who have experienced a relapse [49]. In general, people who had been diagnosed with cancer up to one year ago and those who experienced a relapse of the disease had lower quality of life and higher depression and anxiety scores than those diagnosed more than a year ago and who did not relapse.

The results of this study have shown that the QoL of patients before colorectal tumor resection is influenced by health behaviors and adjuvant therapy. In our opinion, the most significant results are those documenting the influence of health behaviors on quality of life among CRC patients. Our results may help both oncological and patient engagement professionals create educational programs for medical practitioners who can, in turn, directly influence health-oriented behaviors through first-hand contact with patients and their families. Patients expect health professionals to help them change their health behaviors through education, support, treatment, and medication [50,51].

According to the results presented herein, it is worth monitoring behaviors related to health not only in hospitals but also outside medical facilities using strategies such as widely available mobile devices. QoL appraisal including the spectrum of physical, mental, and social levels of functioning may be useful for medical and psycho-oncological personnel when determining interventions. It is worth focusing on all dimensions, both those that have deteriorated and those that are unchanged or improved. Analysis of various dimensions of life may determine a patient’s adaptive abilities. For example, a disease that causes a decrease in a patient’s physical functioning but also an increase in their quality of social functioning may indicate high adaptive ability.

### 4.1. Limitations

Our sample is not representative of the entire country. The majority of respondents were men (roughly 65%). According to the National Cancer Register (KRN), the expected number of men was 55% [52]. Our study group differed from national norms in other ways. For example, 33–36% of our patients suffered from colon cancer, whereas the KRN has indicated that this region accounts for 57% of all large intestinal cancer cases [48]. The number of patients receiving rectosigmoid junction surgery or with rectal cancer was also larger than we had expected.

The inclusion of patients with various cancer locations within the colon and rectum might have affected the results of the study. Another limitation is the fact that this study was conducted with patients who underwent various perioperative therapies and surgical techniques (laparoscopic or open). The adjuvant and neoadjuvant therapy groups were not homogeneous because of their combinations of chemotherapy, radiotherapy, or chemoradiotherapy. However, the composition of the groups (concerning the location of cancer as well as adjuvant and neoadjuvant therapy) in this study at T1 and T2 was similar, and the results provide a picture of the CRC patients.

In our analysis, we focused on the importance of health behaviors with respect to QoL and did not consider some other variables that might be significant. These variables include sociodemographic variables, such as gender, age, and economic status, and psychological variables, such as personality, temperament, emotionality, and cognitive and social systems. For example, personality factors, particularly neuroticism, play a role in lowering scores in quality-of-life assessments [53].

### 4.2. Implications for Research and Practice

Although the importance of physical activity to QoL seems clear, the importance of food consumption for QoL needs further research. A detailed analysis of the impacts of consuming all types of products (healthy and unhealthy) during surgical preparations for convalescence and QoL is necessary.

It is also of interest to conduct research that will highlight the importance of adjuvant therapy for QoL in different types of surgery (laparoscopic hemicolectomy, low anterior rectal resection, and abdominoperineal resection).

Finally, in the future, it is worth conducting studies on more homogeneous groups of patients with CRC.

## 5. Conclusions

In our study, the results revealed that QoL—physical functioning was the area that decreased the most six months after colorectal tumor surgery compared to the period before surgery. Health behaviors such as cessation of smoking, engagement in physical activity, and the consumption of healthy food contribute to higher quality of life among patients before and after colorectal cancer resection. The most significant predictors for QoL—physical functioning were physical activity and the consumption of healthy food. Moderate alcohol consumption had no effect on the quality of life of patients with colorectal cancer.

Patients who received adjuvant/neoadjuvant therapy had lower quality-of-life scores than patients who did not receive this type of therapy. The type of surgery (low anterior rectal resection, laparoscopic hemicolectomy, and abdominoperineal resection) was not related to QoL six months after surgery.

## Figures and Tables

**Figure 1 ijerph-20-05416-f001:**
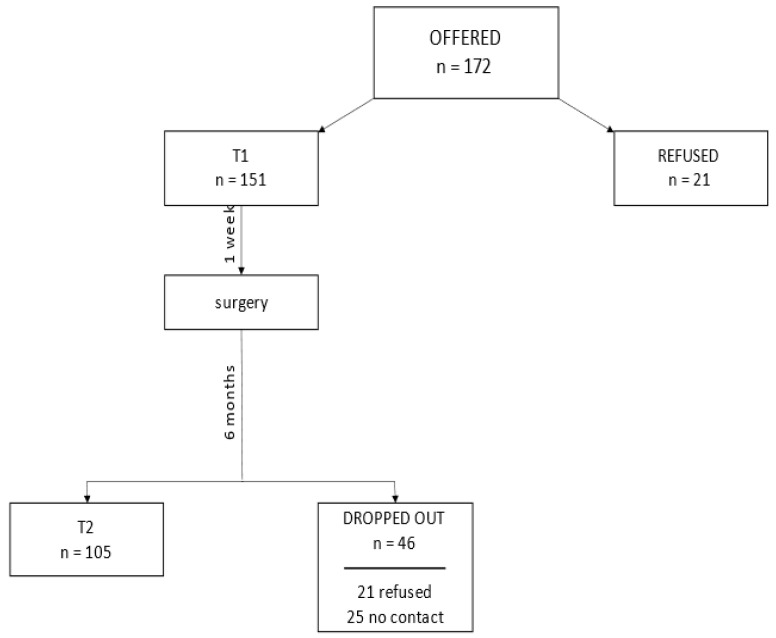
Participant flow.

**Table 1 ijerph-20-05416-t001:** Participants’ characteristics in each sample group before surgery (*n* = 151) and half a year after the surgery (*n* = 105).

	T1 Before Surgery		T2 Half a Year after Surgery	
	Number	%	Number	%
Age (years (SD)) Men (years (SD)) Women (years (SD))	64.89 (10.14) 65.21 (10.73) 64.27 (9.86)		64.30 (10.51) 64.15 (10.20) 64.62 (11.29)	
Gender Men Women	100 51	66.23 33.77	71 34	67.62 32.38
Place of residence: City Country	101 50	66.89 33.11	65 40	61.90 38.10
Marital status Single Married Widowed Divorced	7 115 24 5	4.64 76.16 15.89 3.31	5 82 16 2	4.76 78.10 15.24 1.90
Education Primary Vocational Secondary Higher	22 52 54 23	14.57 34.44 35.76 15.23	12 37 37 19	11.43 35.24 35.24 18.09
Absence of concomitant illnesses Concomitant illnesses	74 77	49.01 50.99	50 55	47.62 52.38
Cancer Colon Rectosigmoid junction Rectum and anal canal Of colon or rectum of uncertain/unknown origin	51 17 73 10	33.77 11.26 48.34 6.62	37 13 48 7	35.24 12.38 45.71 6.67
Neoadjuvant therapy Not applied Chemotherapy Radiotherapy Chemo-radiotherapy	90 2 21 38	59.60 1.32 13.91 25.17	66 2 15 22	62.86 1.90 14.29 20.95
Adjuvant therapy Not applied Chemotherapy Radiotherapy Chemo-radiotherapy	86 56 1 8	56.95 37.09 0.01 5.30	59 43 0 3	56.19 40.95 0.00 2.86
Kind of surgery Laparoscopic hemicolectomy Low rectal anterior resection Abdominoperineal resection	48 69 34	31.79 45.70 22.52	33 52 20	31.43 49.52 19.05
The range of the spread of cancer 0 I II III IV	9 30 42 67 3	5.96 19.87 27.81 44.37 1.99	5 21 33 44 2	4.76 20.00 31.43 49.90 1.90

**Table 2 ijerph-20-05416-t002:** Health behaviors and QoL during T1 and T2.

	T1 (*n* = 151)				T2 (*n* = 105)				Comparing the Change between T1 and T2 F(dt), *p*, η^2^ (*n* = 105)
	M	SD	Min	Max	M	SD	Min	Max	
Number of cigarettes (items/day)	2.12	5.65	0.00	20.00	1.48	5.40	0.00	40.00	F(1,104) = 1.83; *p* = 0.180; η^2^ = 0.02
Alcohol consumption (portions/week)	1.20	4.74	0.00	45.50	0.80	2.40	0.00	14.00	F(1,104) = 2.68; *p* = 0.012.; η^2^ = 0.06
Frequency of consuming healthy food	15.01	3.25	5.00	22.00	15.69	2.64	7.00	21.00	F(1,104) = 0.19; *p* = 0.663.; η^2^ = 0.002
Physical activity (minutes/week)	388.61	308.28	0.00	1680.00	319.43	206.71	0.00	1065.00	F(1,104) = 6.55; *p* = 0.012; η^2^ = 0.06
Global QoL	63.58	19.21	0.00	100.00	67.67	19.62	16.67	100.00	F(1,104) = 1.73; *p* = 0.192.; η^2^ = 0.02
Physical functioning	83.84	16.17	6.67	100.00	80.44	19.13	6.67	100.00	F(1,104) = 10.16; *p* = 0.002.; η^2^ = 0.09
Role-related functioning	85.76	23.76	0.00	100.00	82.06	24.32	0.00	100.00	F(1,104) = 4.86; *p* = 0.029.; η^2^ = 0.05
Emotional functioning	75.37	23.44	0.00	100.00	89.06	12.68	41.67	100.00	F(1,104) = 36.29; *p* < 0.001; η^2^ = 0.26
Cognitive functioning	84.55	20.01	0.00	100.00	86.98	15.67	33.33	100.00	F(1,104) = 2.09; *p* = 0.151; η^2^ = 0.02
Social functioning	86.98	17.64	0.00	100.00	88.57	17.50	33.33	100.00	F(1,104) = 0.32; *p* = 0.571; η^2^ < 0.01
Fatigue	27.08	24.16	0.00	100.00	24.23	21.23	0.00	100.00	F(1,104) = 0.58; *p* = 0.446; η^2^ = 0.01
Pain	19.09	24.97	0.00	100.00	14.29	21.49	0.00	100.00	F(1,104) = 3.00; *p* = 0.086; η^2^ = 0.03

Key: *n*—the size of sample; M—mean value; SD—standard deviation value; Min—minimum value; Max—maximum value; T1—the value of measurement before surgery; T2—the value of measurement half a year after surgery.

**Table 3 ijerph-20-05416-t003:** Quality of life due to the type of surgery and the use of a stoma in T2 (*n* = 105).

		Type of Surgery			The Stoma Application	
		Laparoscopic Hemicolectomy	Low Rectal Anterior Resection	Abdominoperineal Resection	Without a Stoma	Applied Stoma
		*n* = 33	*n* = 52	*n* = 20	*n* = 67	*n* = 38
Global QoL	M (SD)	66.41 (20.46)	68.59 (18.64)	65.83 (21.44)	68.91 (18.49)	64.69 (21.44)
		H(2,105) = 0.31; *p* = 0.857			U = 1122.50; *p* = 0.317	
Physical functioning	M (SD)	79.60 (16.58)	80.64 (20.34)	81.33 (20.70)	80.80 (18.48)	79.82 (20.47)
		H(2,105) = 0.97; *p* = 0.615			U = 1272.50; *p* = 1.00	
Role functioning	M (SD)	79.29 (25.01)	82.69 (24.47)	85.00 (23.51)	82.34 (22.64)	81.58 (27.34)
		H(2,105) = 1.41; *p* = 0.494			U = 1229.00; *p* = 0.772	
Emotional functioning	M (SD)	86.36 (14.85)	90.71 (10.90)	89.17 (12.99)	89.30 (12.96)	88.60 (12.32)
		H(2,105) = 1.54; *p* = 0.463			U = 1228.00; *p* = 0.767	
Cognitive functioning	M (SD)	88.38 (14.72)	86.22 (15.38)	86.67 (18.42)	86.57 (14.86)	87.72 (17.19)
		H(2,105) = 0.41; *p* = 0.816			U = 1165.50; *p* = 0.713	
Social functioning	M (SD)	87.37 (19.11)	87.50 (18.04)	93.33 (12.57)	88.81 (16.76)	88.16 (18.95)
		H(2,105) = 1.77; *p* = 0.413			U = 1263.00; *p* = 0.949	

Key: *n* = sample size; M = mean value; SD = standard deviation; U = Mann–Whitney U test; H = Kruskal–Wallis test; T2—half a year after surgery.

**Table 4 ijerph-20-05416-t004:** The results of the multivariate regression analysis for QoL—physical functioning during T1 and T2.

Variables	T1; R^2^ =.21; R^2^ Adjusted = 0.18 F(5,142) = 7.52; *p* < 0.001 VIF = 5.34					Variables	T2; R^2^ = 0.27; R^2^ Adjusted = 0.22 F(5,93) = 6.90; *p* < 0.001 VIF = 5.70				
	*β*	SE *β*	t	*p*	R^2^		*β*	SE *β*	t	*p*	R^2^
Absolute term			9.82	<0.001		Absolute term			5.48	<0.001	
Number of cigarettes (items/day)	−0.18	0.08	−2.34	0.020	0.04	Number of cigarettes (items/day)	−0.13	0.10	−1.29	0.200	0.02
Alcohol intake (portions/week)	0.08	0.08	1.12	0.266	<0.01	Alcohol intake (portions/week)	0.06	0.10	0.58	0.566	<0.01
Frequency of consuming healthy food	0.22	0.08	2.81	0.006	0.04	Frequency of consuming healthy food	0.19	0.09	2.02	0.046	0.03
Physical activity (minutes/week)	0.26	0.08	3.37	<0.001	0.10	Physical activity (minutes/week)	0.41	0.09	4.60	<0.001	0.19
Neoadjuvant therapy	−0.18	0.08	−2.54	0.012	0.03	Adjuvant therapy	−0.19	0.09	−2.09	0.039	0.02

Key: T1—the value before colon resection; T2—the value six months after surgery; R^2^—the coefficient of determination; Adjusted R^2^—the corrected coefficient of determination; *β*—a standardized regression coefficient for T1 or T2; SE *β*—the standard error for the standardized beta; VIF—Variance Inflation Factor.

**Table 5 ijerph-20-05416-t005:** The results of the multivariate regression analysis for QoL—social functioning during T1 and T2.

Variables	T1; R^2^ =.28; R^2^ Adjusted = 0.25 F(5,138) = 10.52; *p* < 0.001 VIF = 5.25					Variables	T2; R^2^ = 0.24; R^2^ Adjusted = 0.20 F(5,94) = 6.10; *p* < 0.001 VIF = 5.44				
	*β*	SE *β*	t	*p*	R^2^		*β*	SE *β*	t	*p*	R^2^
Absolute term			14.75	<0.001		Absolute term			11.57	<0.001	
Number of cigarettes (items/day)	−0.36	0.08	−4.81	<0.001	0.11	Number of cigarettes (items/day)	−0.34	0.10	−3.47	<0.001	0.08
Alcohol intake (portions/week)	0.10	0.08	1.42	0.159	0.01	Alcohol intake (portions/week)	0.14	0.10	1.44	0.154	0.02
Frequency of consuming healthy food	0.33	0.08	4.32	<0.001	0.09	Frequency of consuming healthy food	0.31	0.10	3.41	<0.001	0.10
Physical activity (minutes/week)	0.19	0.08	2.62	0.010	0.04	Physical activity (minutes/week)	0.21	0.09	2.34	0.021	0.04
Neoadjuvant therapy	−0.18	0.08	2.39	0.018	0.03	Adjuvant therapy	−0.02	0.09	−0.22	0.823	<0.01

Key: T1—the value before colon resection; T2—the value six months after surgery; R^2^—the coefficient of determination; Adjusted R^2^—the corrected coefficient of determination; *β*—a standardized regression coefficient for T1 or T2; SE *β*—the standard error for the standardized beta; *VIF*—Variance Inflation Factor.

**Table 6 ijerph-20-05416-t006:** The results of the multivariate regression analysis for QoL—emotional functioning during T1 and T2.

Variables	T1; R^2^ =.22; R^2^ Adjusted = 0.20 F(5,145) = 8.31; *p* < 0.001 VIF = 5.25					Variables	T2; R^2^ = 0.16; R^2^ Adjusted = 0.12 F(5,97) = 3.82; *p* < 0.001 VIF = 5.55				
	*β*	SE *β*	t	*p*	R^2^		*β*	SE *β*	t	*p*	R^2^
Absolute term			8.94	<0.001		Absolute term			22.15	<0.001	
Number of cigarettes (items/day)	−0.38	0.08	−5.03	<0.001	0.09	Number of cigarettes (items/day)	−0.31	0.10	−3.00	0.003	0.03
Alcohol consumption (portions/week)	0.09	0.08	1.26	0.210	<0.01	Alcohol consumption (portions/week)	0.20	0.10	2.01	0.047	0.03
Frequency of consuming healthy food	0.29	0.08	3.81	<0.001	0.09	Frequency of consuming healthy food	0.21	0.09	2.22	0.029	0.05
Physical activity (minutes/week)	0.05	0.08	0.71	0.480	<0.01	Physical activity (minutes/week)	0.01	0.09	0.05	0.957	<0.01
Neoadjuvant therapy	−0.19	0.08	2.57	0.011	0.04	Adjuvant therapy	−0.21	0.10	−3.00	0.003	0.05

Key: T1—the value before colon cancer resection; T2—the value six months after surgery; R^2^—the coefficient of determination; Adjusted R^2^—the corrected coefficient of determination; *β*—a standardized regression coefficient for T1 or T2; SE *β*—the standard error for the standardized beta; *VIF*—Variance Inflation Factor.

## Data Availability

Not applicable.

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
