# Peer review of "The Role of Health Behaviors in Quality of Life: A Longitudinal Study of Patients with Colorectal Cancer"

_ijerph, 2023, doi:10.3390/ijerph20075416_

Round 1
Reviewer 1 Report
I have reviewed the manuscript, titled “The role of health behaviors for quality of life: A longitudinal study of patients with colorectal cancer”. The study was to investigate the role of health behavior and quality of life (QoL) among patients with CRC receiving surgical treatment.
The study has some strong points (clear introduction, accurate statistical analysis, constructive discussion).
However, I would like to ask the authors to address some points in order to improve the paper
Introduction:
1) As the study examines health behavior and quality of life in cancer patients, it would be appropriate to provide more information on the role of quality of life and well-being among cancer patients (see: Triberti, S., Savioni, L., Sebri, V., & Pravettoni, G. (2019). eHealth for improving quality of life in breast cancer patients: a systematic review. Cancer Treatment Reviews, 74, 1-14; Krok, D., Telka, E., & Zarzycka, B. (2022). Modeling psychological well‐being among abdominal and pelvic cancer patients: The roles of total pain, meaning in life, and coping. Psycho‐Oncology, 31(11), 1852-1859).
2) Can you present some explanations for relationships between health behavior and quality of life? (p.2)
Method:
3) Was the sample determined by power analysis?
4) How did you handle missing values in your data? (If any exist)
Results:
5) This section is excellently presented, so I do not have any suggestions.
Discussion:
6) What are the underlying mechanisms responsible for this result: “Emotional functioning was related to smoking, consumption of healthy food, and complementary treatments” (p. 11)?
7) Can you elaborate on the following statement: “No changes of statistical significance were found in the consumption of healthy food before or six months after surgery. (p. 12).”. Please, provide a potential explanation of this result.
Author Response
Dear Reviewer, thank you very much for your time and critical evaluation of the manuscript.
We have addressed your comments below.
Introduction:
- As the study examines health behavior and quality of life in cancer patients, it would be appropriate to provide more information on the role of quality of life and well-being among cancer patients (see: Triberti, S., Savioni, L., Sebri, V., & Pravettoni, G. (2019). eHealth for improving quality of life in breast cancer patients: a systematic review. Cancer Treatment Reviews, 74, 1-14; Krok, D., Telka, E., & Zarzycka, B. (2022). Modeling psychological well‐being among abdominal and pelvic cancer patients: The roles of total pain, meaning in life, and coping. Psycho‐Oncology, 31(11), 1852-1859).
- Can you present some explanations for relationships between health behavior and quality of life? (p.2).
We have supplemented the introduction with the following section using suggested references:
Quality of life refers to an individual’s subjective assessment of their life situation. In the case of cancer patients, their health condition is an important factor for their quality of life. Symptoms of the disease, such as pain, have a negative impact on their psychological well-being and quality of life [15]. Studies indicate that coping strategies mediate the relationship between pain and well-being [16,17]. Coping can involve emotional strategies like relaxation, distraction, or meditation, as well as behavioral strategies such as engaging in physical activity, adopting a healthy diet, or using stimulants like smoking or drinking alcohol.
In addition, there is a section in the introduction describing the relationship between health behavior and quality of life
Studies have shown that health behaviors are related to QoL. In patients with CRC, an improvement in QoL related to physical functioning is observed as a result of increased physical activity and proper diet [18]. Health behaviors that favor increased QoL are consumption of vegetables and fruits [19,20] and physical activity [21,22]. Studies also emphasize the negative impact of smoking and drinking alcohol on QoL [20,23].
Method:
- Was the sample determined by power analysis?
We did not perform power analysis. At the research design stage, we consulted statisticians who defined the number of 100 surveys as the minimum number to perform a good quality statistical analysis. The number of 100 participants was determined by taking into account the statistical tests they planned to use in the analysis of the results
4) How did you handle missing values in your data? (If any exist)
There were no missing data. The research was conducted by interviewers who cared about the quality of data.
Discussion:
6) What are the underlying mechanisms responsible for this result: “Emotional functioning was related to smoking, consumption of healthy food, and complementary treatments” (p. 11)?
In the discussion, the relationship between smoking and emotional functioning was clarified.
Research confirms that even in the general population, cigarette smokers present lower QoL than non-smokers and smoking cessation improves QoL [34–36]. Many mechanisms impact this effect, but within the scope of physical performance, a decrease in oxygen caused by smoking reduces physical endurance, making it more difficult to perform well in sports and everyday activities such as walking up stairs. The impact of smoking on emotional functioning can be explained by the nicotine contained in cigarettes, which penetrates the central nervous system, causing sensations such as relaxation, increased concentration, or improved mood. Conversely, limiting or abandoning smoking by cigarette smokers is associated with negative feelings such as increased anxiety, mood deterioration, and possibly increased fear [37,38]. This may partially why there is lower emotional functioning among people who smoke but try to limit the number of cigarettes.
We added a section about the relationship between healthy food consumption and emotional functioning.
The relationship between consuming healthy food and improved emotional functioning can be characterized as the "food-mood connection mechanism." This suggests that a person who consumes a particular food product combines input from numerous receptors (e.g. taste, smell, visual, auditory) with their current emotional state during a single experience. Through repeated experiences, an individual's association between healthy foods and positive emotional states is strengthened [41].
The description of relationship between adjuvant/neoadjuvant therapy and emotional functioning has been completed and is presented as follows:
This study also revealed a relationship between adjuvant/neoadjuvant therapy and decreased QoL. Perioperative treatment (preoperative/postoperative radio- and chemotherapy) plays a crucial role in contemporary treatment of CRC patients but might have an important impact on QoL. Preoperative radiotherapy is a risk factor for low anterior resection syndrome, which significantly reduces quality of life [45]. Similarly, radiotherapy significantly increases sexual dysfunction [46] and impairs urination, particularly with distal tumors treated with abdominoperineal resection [47].
The emotional functioning of patients who have undergone additional adjuvant treatment is at a lower level compared to those who have not.
7) Can you elaborate on the following statement: “No changes of statistical significance were found in the consumption of healthy food before or six months after surgery. (p. 12).”. Please, provide a potential explanation of this result.
Table 2 presents the values of eating healthy food before the operation and half a year later. It shows that there are no statistically significant changes in consumption of healthy food at these two time points.
However, this variable differentiates patients in terms of physical, social and emotional functioning.
Reviewer 2 Report
Important focus for : Colorectal cancer patients, which explored the QOL of patients receiving surgical treatment. Well followed the methodology and articulated the significance of the study. Could modify the result section by not being duplicating the information provided in the table and having data in the table form than being descriptive. Well written discussion and drawn recommendations.

Author Response
Dear Reviewer, thank you very much for your time and critical evaluation of the manuscript and your comment below.
Could modify the result section by not being duplicating the information provided in the table and having data in the table form than being descriptive.
Referring to the issue of the presentation of results and the dilemma whether to describe the results visible in the tables and figures or not.
We decided to describe the presented results due to the ease of reading such an article and the lack of restrictions on the number of words used in an open access journal.
Looking through the literature, you can find the presentation of the results proposed by the P.T. reviewer as well as the one we have made.
We would prefer to keep it that way, but if absolutely necessary we will reduce the description of the results section.